# Limited evidence that body size shrinking and shape-shifting alleviate thermoregulatory pressures in a warmer world

Joshua K. R. Tabh [1]✉, Elin Persson [1], Maria Correia [1,2], Ciarán Ó Cuív [1,3], Elisa Thoral [1,4] & Andreas Nord [1,5]

Amassing evidence indicates that vertebrates across the globe are shrinking and changing shape concurrent with rising temperatures. Ecogeographical theories assert that these changes should provide thermoregulatory benefits by easing heat dissipation, however, thermophysical models underpinning such theories are highly simplified and lack empirical validation. Using data from three temperature-manipulation experiments, we quantified the contributions of body size and appendage lengths toward thermoregulatory performance in Japanese quail, while simultaneously querying neutral plasticity as an alternative driver of avian shape-shifts. In the cold, body mass and leg length (here, tarsus length) influenced energy costs of warming, but only among juveniles. In the warmth, smaller body sizes, longer legs and longer bills independently reduced energy and water costs of cooling across ages, but whole-body phenotypes necessary to provide even moderate thermoregulatory benefits were rare (2.5%) and required large departures from allometry. Last, rearing in the warmth reduced body sizes and increased appendage lengths comparable to recent changes observed in nature, but emergent morphologies provided no clear thermoregulatory benefit. Our findings question whether shrinking and shape-shifting are indeed easing thermoregulation in birds or reflect selection for such. Neutral plasticity, or relaxed selection against small body size in juveniles, may better explain recent avian shape-shifting.

In 2022, 'Dippy', the cast replica of a *Diplodocus carnegii* skeleton unearthed in the late 1800s, returned for display at the United Kingdom's Natural History Museum after several years of absence. Within 6 months of exposure, over 1 million attendants sought to view the cast, rendering Dippy the museums 'most popular exhibition' of the year[1]. That Dippy stands an impressive 6 metres tall and 26 metres long is undoubtedly one of the keys to its allure; we are fascinated with large-bodied animals[2,3]. Yet opposing this fascination, amassing evidence indicates that a wide range of extant bird and mammal species are shrinking and changing shape[4–7]. How or why these 'shape-shifts' have occurred is not yet known, however, given a concurrence with rising global temperatures, many have interpreted them as beneficial for thermoregulation[6,8,9] and potentially driven by selection or adaptive

plasticity. By decreasing body size and adjusting shape, surface area to volume ratios may increase, thus easing heat dissipation in a warmer world.

The logic behind thermoregulation as a driver for species' shape-shifts is not novel. Over a century ago, Bergmann[10] and Allen[11] applied the same logic to explain their observations that body size generally decreases, and appendage length increases, with increasing environmental temperatures across the globe (phenomena known as Bergmann's and Allen's rules, respectively). Despite this, whether body size and shape do have meaningful impacts on costs of thermoregulation in endotherms with evolved heat-retention and dissipation mechanisms is tenuous. Already 70 years ago, for example, Scholander[12] argued that low surface area to volume ratios should be irrelevant for endotherm survival in the cold since heat loss is so readily

[1]Department of Biology, Lund University, Lund, Sweden. [2]Department of Biological and Environmental Science, University of Jyväskylä, Jyväskylä, Finland. [3]Department of Wildlife, Fish and Environmental Studies, Swedish University of Agricultural Sciences, Umeå, Sweden. [4]Département de Biologie, LIttoral ENvironnement et Sociétés (LIENSs), La Rochelle Université, La Rochelle, France. [5]Swedish Centre for Impacts of Climate Extremes (climes), Lund University, Lund, Sweden. ✉e-mail: joshua.tabh@biol.lu.se

mitigated via insulation, heat exchangers, and peripheral vascular constriction (supported by ref. 13). In the heat, predictions from physiological allometries across species have raised similar doubts about the importance of morphology for thermoregulatory costs[14]. Yet even with such long-standing theoretical discussion, remarkably little is empirically known about whether body size and shape do directly influence costs of thermoregulation within a species. One probable hinderance is that adult morphometry and thermal physiology are often part of an interlaced phenotype that is shaped by temperature experienced during development[15–17]. Teasing out any direct effects of morphology on thermoregulation is thus empirically challenging, and probably explains mixed conclusions of the few studies striving to do so[18–22]. Still, if we wish to determine whether shape-shifting in extant endotherm species is driven by selection on, or adaptive plasticity for, improved thermoregulatory efficiency in a warming world, direct effects of morphology on thermoregulation must first be understood.

In this study, we used a line of open-air breeding Japanese quail (*Coturnix japonica*) to test whether body size (here, body mass, a strong predictor of skeletal size in this species; see 'Methods') and appendage lengths (here, tarsus and bill lengths) have a direct influence on costs of thermoregulation in the heat and cold. The Japanese quail was chosen as our model for its precociality, allowing us to control for early thermal experience by rearing individuals at fixed ambient temperatures without bias from parental brooding. Given that recent size declines and shape-shifts have been reported for both precocial[23,24] and altricial species[5,8], our findings are expected to be generalisable across life-history groupings. To measure thermoregulatory costs, we quantified the rate at which individuals increased their resting metabolism across moderate cooling (from 30 °C to 10 °C) and moderate heating (from 30 °C to 40 °C) events (here, defined as 'metabolic slopes'); in this way, more rapid rates of increase (i.e. 'steeper' metabolic slopes) indicate higher energy costs of thermoregulation than slower rates of increase. To better understand mechanisms linking morphology to thermoregulation in the heat, these measurements were supplemented with those of evaporative cooling efficiency (the ratio of evaporative heat loss to metabolic heat production) at temperatures limiting sensible heat loss (40 °C). Effects of body mass and appendage lengths on all measures of thermoregulatory costs were evaluated using Bayesian regressions. Last, rearing conditions of quail were varied (10 °C [cold], 20 °C [mild], and 30 °C [warm]) and both metabolic slopes and growth curves compared across treatments to: (1) contrast the importance of morphological contributions to thermoregulation against those of physiological acclimation, and (2) evaluate neutral phenotypic plasticity as an alternative driver of modern avian shape-shifts. All analyses were carried out on juvenile (3-week-old) and adult quail (8-week-old, the age of reproductive maturity[25]).

## Results and discussion
Bayes Factors (BF) are given for all test statistics (e.g. model coefficients, or βs) in place of credible intervals and represent relative support for the alternative hypothesis over the null. A BF of ≥3 represents moderate support for an alternative hypothesis, with corresponding 50% credible intervals not crossing 0. Credible intervals (50% and 95%) are provided in the Supplementary Information.

### Body size and appendage length weakly influence thermoregulatory costs in the cold, and only in juveniles
Large bodies and short appendages are widely assumed to reduce the costs of thermoregulation in the cold by decreasing surface area to volume ratios and thus lowering rates of sensible heat loss. Among adult birds, however, our findings do not support this assumption. After maturity, neither body mass, tarsus length, nor bill length influenced metabolic slopes below thermoneutrality (Fig. 1A–C; n = 52; mass: β ≈ 0, BF = 1.481, partial $R^2$ = 0.014 [95%: 0, 0.173]; tarsus length: β ≈ 0, BF = 1.656; partial $R^2$ = 0.016 [95%: 0, 0.172]; bill length: β ≈ 0, BF = 1.548; partial $R^2$ ≈ 0 [95%: 0, 0.167]; Supplementary Tables 52 and 53), despite each trait varying widely among individuals (coefficients of variation [C.V.s]: mass = 13.3%, tarsus

length = 6.4%, bill length = 8.7%; for comparison, average C.V.s for body mass and tarsus length across 54 avian species [ref. 5] = 9.2% and 3.5% respectively). Even when morphology was dramatically skewed to increased surface area (i.e. a mass 2 standard deviations [SDs] below the mean, or appendage lengths 2 SDs above the mean), predicted metabolic responses to the cold were nearly identical to those of average-shaped individuals (differences in metabolic slopes at: [1] average—atypically small mass ≈ 0, BF = 1.11, [2] average—atypically long tarsus length ≈ 0, BF = 1.13, [3] average—atypically long bill length ≈ 0, BF = 1.00). That our experiment controlled for life-time temperature experience indicates that mere variability in thermal acclimation is not sufficient to explain previous failures to link morphology with cold-induced metabolic responses in wild birds[18,22]. Instead, a consistent failure to do so likely indicates a sufficiency of physiological heat-retention mechanisms (e.g. peripheral vasoconstriction[12,26]) that are able to offset the added heat-loss risks of a high body surface area.

An alternative explanation for why we did not detect effects of adult morphometry on metabolic responses to cold is that metabolic slopes were either highly variable within individuals (i.e. 'noisy') or highly similar among individuals. In these cases, thermoregulatory phenotypes would be too indistinguishable to detect any influence of morphometry on them. However, similar to other thermo-physiological traits (e.g. cold-induced maximal metabolism and peripheral vasomotor responses[27,28]), metabolic slopes were indeed distinct among our quail (Fig. 1D; conditional repeatability = 0.348 [95%: 0.169, 0.560]; probability of exceeding repeatability in null model ≈ 100%; Supplementary Fig. 90), but neither body size nor appendage lengths explained their distinctions. Alternatively, larger bodies and shortened appendages may still have benefited thermoregulation by either: (1) reducing risk of hypothermia, or (2) lowering the temperature at which metabolism must be increased to stabilise body temperature (i.e. by decreasing lower critical temperature[29]). Our data, however, provided little evidence to support these alternatives as well. Among the majority of adults (87.5%), body temperature increased following cold exposure rather than decreasing, and the direction and extent to which body temperature changed in the cold did not vary by morphology (see Supplementary Table 117; body temperature responses to the cold: body mass: β = −1.0 × 10$^{-4}$, BF = 1.057; tarsus length: β = −1.53 × 10$^{-2}$, BF = 2.336; bill length: β = −0.037, BF = 1.877; n = 51). Further, of all morphological traits, only bill length influenced lower critical temperature, and the size of this effect was limited (corresponding to a 0.08 °C decrease in lower critical temperature with every SD decrease in bill length; bill length effect: β = 0.143, BF = 4.481; body mass effect: β = −1.8 × 10$^{-3}$, BF = 2.225; tarsus length effect: β = 0.033, BF = 1.908; Supplementary Table 114).

Among juveniles, effects of body mass and tarsus length on metabolic slopes were clear (body mass: β = 2.0 × 10$^{-4}$, BF = 27.571; partial $R^2$ = 0.082 [0, 0.276]; tarsus length: β = −4.0 × 10$^{-4}$, BF = 3.075; partial $R^2$ = 0.024 [0, 0.225]; n = 42; Supplementary Tables 41 and 42) while those of bill length were still absent (β = 4.0 × 10$^{-4}$, BF = 2.085; partial $R^2$ = 0.002 [0, 0.232]; Supplementary Tables 41 and 42). Together, these effects largely signalled compensation for increased rates of heat loss when surface area to volume ratios were highest—a prediction of long-standing, biophysical models (i.e. Bergmann's and Allen's rules). More specifically, birds with smaller body masses and longer tarsi increased their resting metabolism slightly more in response to the cold relative to those with larger body masses and shorter tarsi (Supplementary Fig. 95). In Japanese quail, plumage development remains incomplete until near sexual maturation (3 weeks after our juvenile measurements[30]) and cold-induced vasoconstriction—particularly at the limbs—is likely limited during growth owing to demands for micro- and macro-nutrient delivery[31]. That early juveniles appear more influenced by body size and limb length scaling than adults may therefore be unsurprising. With respect to recent avian shape-shifts, these age-specific effects of body size and limb length on cold-induced metabolism could help explain their occurrence, if warming temperatures are relaxing selection against small body size and limb length during development[32]. However, effect sizes from our model indicated that both the influence of body mass and tarsus length on metabolic responses to cold were small (average mass vs. average

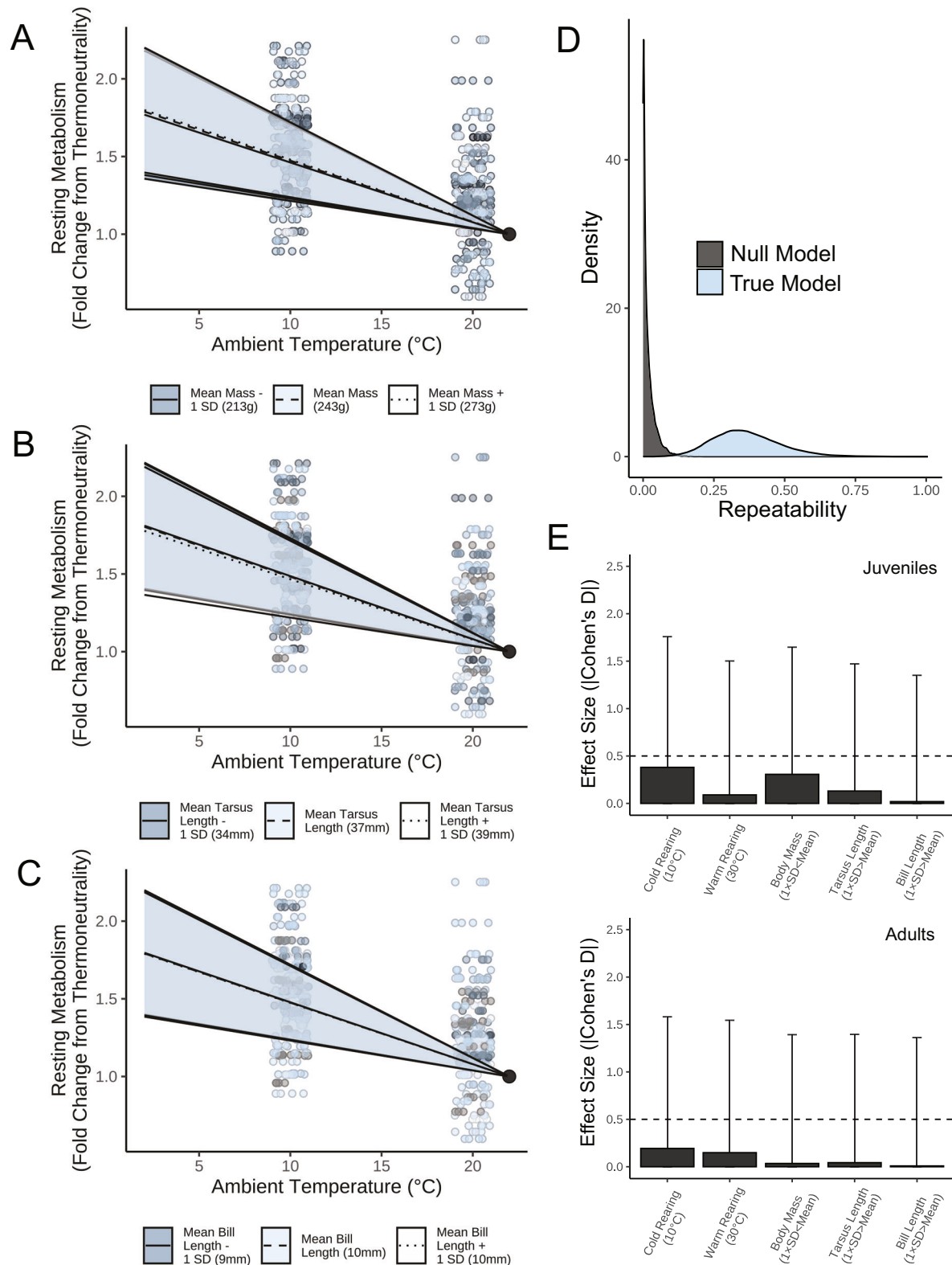

**Fig. 1 | Contributions of morphology, individual identity, and prior temperature experience on metabolic responses to cold in Japanese quail (n = 52 adults; n = 42 juveniles). A–C** Effects of body mass, tarsus length and bill length on metabolic responses to cold in adults. Resting metabolism is relativised by individual to represent their fold change from thermoneutrality (here, 30 °C). Lines and ribbons represent predicted effects ±1 standard error, respectively, holding other variables constant. Small dots represent raw values, with colour scaled by relative morphological size. **D** Conditional repeatabilities of adult metabolic slopes below thermoneutrality (calculated following Schielzeth and Nakagawa[74]). 'True model' indicates repeatability estimates derived from a model where individual identities were known and correctly labelled; 'Null' model indicates repeatability estimates derived from a model where individual identities were randomly scrambled. **E** Absolute effect sizes (here, Cohen's D) of given predictors on metabolic slopes of juvenile and adult quail. Bars display means and error bars indicate standard deviations.

mass + 1 SD: Cohen's D = 0.31; average tarsus length vs. average tarsus length + 1 SD: Cohen's D = 0.13; Fig. 1E), with atypically small body masses (2 SDs below the mean) and atypically long tarsi (2 SDs above the mean) predicted to increase metabolism in the cold (10 °C) by only 6.5% and 2.8% above average. Whether these effect sizes are sufficient to influence patterns of selection (i.e. by easing survivorship of small-bodied individuals in increasingly milder temperatures[32,33]) is not clear but should be investigated to fully understand the contribution of thermoregulatory costs toward shifts in morphology in several extant bird species.

Similar to morphology, we found that thermal history directly influenced metabolic responses to the cold, but only among juveniles. Here, cold rearing (10 °C) reduced the extent to which metabolism increased below thermoneutrality (juveniles; $\beta = 3.6 \times 10^{-3}$, BF = 6.117; adults: $\beta = -3.8 \times 10^{-3}$, BF = 2.524; Supplementary Tables 41 and 52) while warm rearing had no clear effect on this increase (juveniles: $\beta = -9.0 \times 10^{-4}$, BF = 1.341; adults: $\beta = 3.1 \times 10^{-3}$, BF = 2.012; Supplementary Tables 41 and 52). By comparison, effects of cold rearing on metabolic responses exceeded those of relatively substantial changes in body mass and appendage lengths (i.e. an increase or decrease in each trait by 1 SD; Fig. 1E). While the mechanisms driving this effect are not clear, developing at low ambient temperatures may have enhanced efficiency of heat storage in our quail[34,35] (but see ref. 36), either by promoting plumage growth[37,38], or modifying rates of blood flow at the periphery[39]. Given the comparatively large size of this acclimation effect relative to that from morphology, it appears unlikely that selection pressures from low ambient temperatures necessarily drives changes in morphology. Indeed this conclusion was already reached nearly 70 years ago, when Per Scholander argued that physiological acclimation is probably sufficient to compensate for added heat loss costs of high surface area to volume ratios[12].

## Body size and appendage length influence thermoregulation in the heat, but in opposite directions

Ecogeographical rules describing temperature-size relationships across endotherms generally focus on effects of low ambient temperatures on morphology, rather than high[10,11]. However, by physical principles, warm environments should also favour smaller and longer-limbed animals owing to their larger surface area to volume ratios and expectedly higher sensible heat dissipation rates. We tested this assumption by evaluating whether and how body mass and appendage lengths influenced the extent to which metabolism of quail (n = 80) increased during a heat challenge (40 °C; ~10 °C above the upper critical temperature for this species[40]).

As predicted by size-related heat dissipation capacities, the rate at which adult birds increased their metabolism in the heat increased with body mass (Fig. 2A; $\beta = 3.0 \times 10^{-4}$, BF = 66.227; Supplementary Table 79), although with limited explanatory power (partial $R^2$ = 0.046 [95%: 0, 0.169]; Supplementary Table 80). Trends in evaporative cooling efficiency (i.e. the fraction of total heat production lost by evaporation) suggest that elevated metabolic costs in large birds represented a reduced capacity to counteract metabolic heat production with evaporative heat loss (effect of mass on evaporative cooling capacity: $\beta = -1.4 \times 10^{-3}$, BF = 23.465; n = 61; Supplementary Table 103), the dominant form of heat dissipation at ambient temperatures near, or above, body temperature[41]. Despite this, body temperature responses to the heat did not vary across body masses ($\beta = 1.0 \times 10^{-3}$; BF = 2.047; n = 80; Supplementary Table 120) indicating no added risk of hyperthermia in larger individuals. Among atypically heavy birds (i.e. mean mass + 2 SDs), added metabolic costs of heat exposure were predicted to be large, with relative metabolism increasing by ~27.9% from thermoneutrality compared with a modest 8.7% among average-massed birds (holding tarsus and bill lengths constant). These findings imply that a large body size impedes thermoregulatory performance in a warmer world, as assumed by many (*sensu* refs. 4,8,9).

At the level of the appendages, metabolic costs of heat exposure marginally decreased with increasing length (Fig. 2B, C; tarsus length: $\beta = -2.4 \times 10^{-3}$, BF = 7.762, partial $R^2$ = 0.025 [0, 0.147]; bill length: $\beta = -5.6 \times 10^{-3}$, BF = 3.988, partial $R^2 \approx 0$ [0, 0.153]; Supplementary

Tables 79 and 80), also as predicted by biophysical theory. However, in our population, tarsus length (but not bill length) displayed clear, positive allometry with body mass (tarsus length: $\beta = 0.029$, BF > 1000; bill length: $\beta = -4.0 \times 10^{-4}$, BF = 1.277). As such, metabolic benefits accrued by individuals with relatively long tarsi (here, by increasing sensible heat loss; see no effect of tarsus length on evaporative cooling efficiency: $\beta = 4.6 \times 10^{-3}$, BF = 1.967, partial $R^2$ = 0.011 [0, 0.197]; Supplementary Tables 103 and 105) were largely negated by costs of their comparatively larger mass. Supporting this, whole-body phenotypes leading to moderate (Cohen's D ≤ −0.5) or large (Cohen's D ≤ −0.8) metabolic benefits in the heat required extreme deviations from allometry which were absent in our population, while those leading to moderate or large metabolic costs (Cohen's D ≥ −0.5 or −0.8 respectively) were rare (moderate effects = ~2.5%, large effects = 0%; Fig. 2D). Being uncorrelated with body mass, bill size alone remained an independent, morphological affector of metabolic costs in the heat, although again with little consequence to whole-body metabolic costs (Fig. 2C, E; see above). While the precise mechanisms linking bill length with reduced metabolic expenditure in the heat is not evident here, an elevated evaporative cooling efficiency in longer-billed quail ($\beta = 0.049$, BF = 9.610, partial $R^2$ = 0.030 [0, 0.218]; Supplementary Tables 103 and 105) appears to indicate an increased capacity to dissipate heat through wet, rather than dry, (or 'sensible') means among these individuals (possibly owing to a larger surface area of wet tissues).

When we analysed heat-induced metabolic responses in juveniles, moderate effects of body mass and appendage lengths again emerged. Consistent with adults and predictions from biophysical principles, larger juveniles with both shorter tarsi and bills tended to increase their metabolism more in the heat than smaller juveniles with longer tarsi and bills (mass: $\beta = 5.0 \times 10^{-4}$, BF = 15.194; partial $R^2$ = 0.015 [0, 0.136]; tarsus length: $\beta = -1.0 \times 10^{-3}$, BF = 3.206; partial $R^2$ = $4.0 \times 10^{-3}$ [0, 0.126]; bill length: $\beta = -0.011$, BF = 19.942; partial $R^2$ = 0.030 [0, 0.142]; n = 66; Supplementary Tables 69 and 70). Again, higher metabolic responses among heavier juveniles appeared to be a consequence of their comparatively weaker capacity to dissipate heat evaporatively (effect of body mass on evaporative cooling capacity: $\beta = -2.60 \times 10^{-3}$, BF = 141.857; Supplementary Table 94). Nevertheless, given that large individuals generally had longer tarsi ($\beta = 0.064$, BF > 1000; but not larger bills: $\beta = 3.8 \times 10^{-3}$, BF = 2.882) that offset these costs, whole-body phenotypes displaying moderate (Cohen's D ≥ 0.5) or high (Cohen's D ≥ 0.8) increases in metabolism relative to average remained rare at this life stage (6% for moderate increases and 0% for large effects).

The above findings provide tentative evidence that shifts in body size and appendage length could benefit endotherm thermoregulation in a warming world. Exactly how and to what degree these benefits might shape selection in nature, however, is not obvious (see ref. 42). That effects of body mass and tarsus length on metabolic responses to heat evidently counteract each other indicates that modifying allometries between these traits, rather than altering each trait individually, is first required for selective benefits to occur. Supporting this, the relationship between body size and appendage length appears to explain thermal niche across species better then each trait individually[43]. In our study, however, deviations from allometry between body size and tarsus length that led to at least moderate thermoregulatory costs were uncommon (~2.5%; discussed above), and removal of these individuals (simulating selective disappearance) had no clear effect on the relationship between tarsus length and body mass across our population (change in slope = $6.61 \times 10^{-3} \pm 0.011$; BF = 2.962; holding all other effects constant). Consequences of selection against these particular extremes may therefore by negligible for population-level phenotypes. At the level of the bill, however, allometric constraints on size were either absent or too weak to detect in our population. This relative independence, coupled with marginal effects of bill length on metabolic responses to heating, suggests that bill size may be more free to respond to selection for thermoregulatory efficiency than the tarsi, barring functional costs of enlargement (for example, on preening[44] and foraging[45]). Whether an effect of bill length on metabolic responses to heating is large enough to both facilitate such selection and

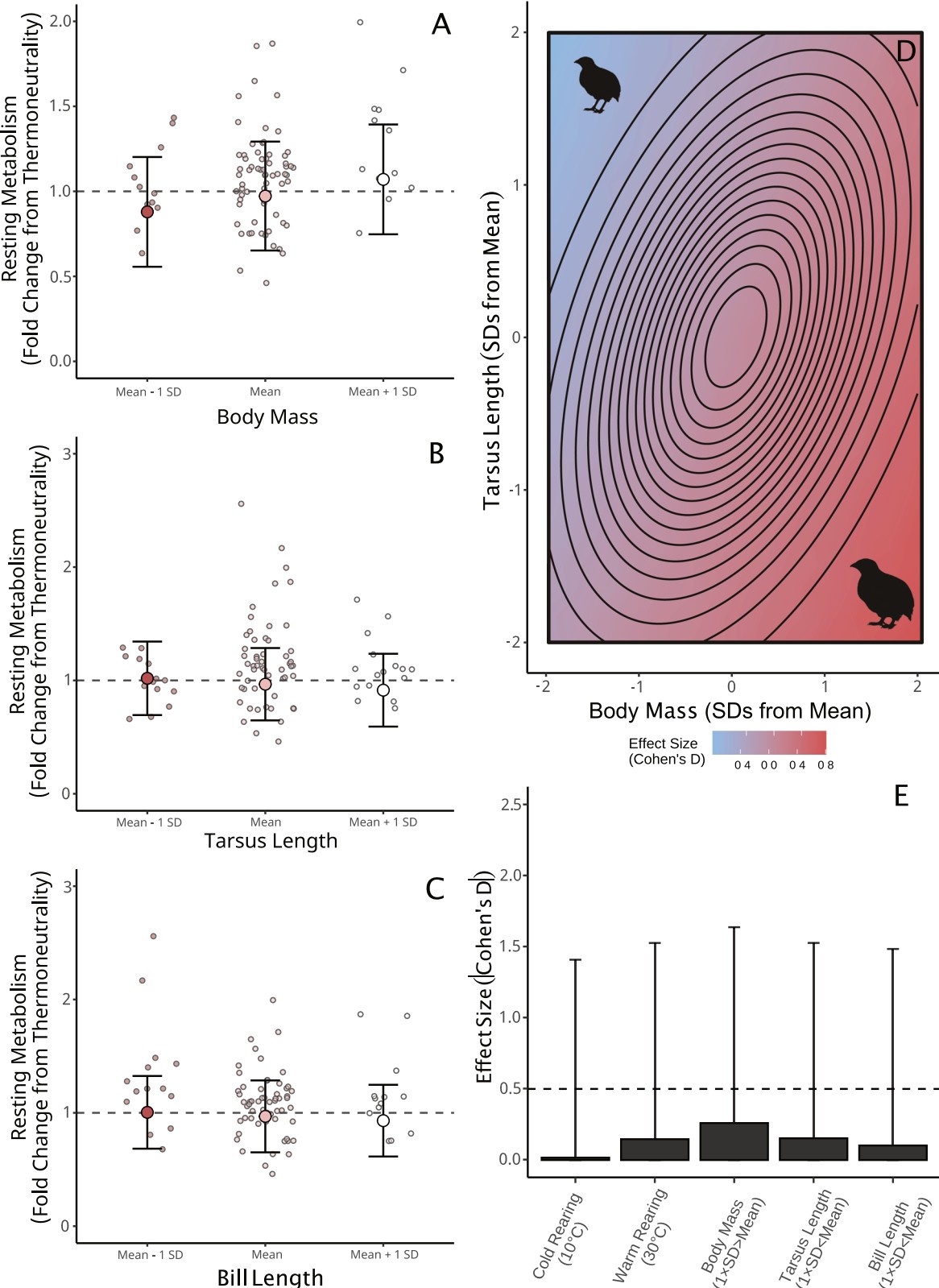

**Fig. 2 | Contributions of morphology and prior temperature experience on metabolic responses to heat in adult Japanese quail (n = 80). A–C** Effects of body mass, tarsus length, and bill length on resting metabolic rate, with resting metabolism relativised to represent an individual's fold change from thermoneutrality (here, 30 °C). Large dots indicate predicted effects, holding other variables constant. Error bars indicate ±1 standard deviation around predictions. Small dots display raw values. **D** The combined effects of body mass and tarsus length on metabolic slopes in the heat relative to the mean. Effects are shown as mean Cohen's D values. Contours show the multivariate, normal distribution of phenotypes among quail. The centre circle indicates the modal phenotype for our population (i.e. the peak of the distribution). Outer circles represent the tails of the distribution, where phenotypes within are rare. **E** Absolute effect sizes (here, Cohen's D) of given predictors on metabolic slopes. Bars display means and error bars indicate standard deviations. The quail silhouette was created by J.T.

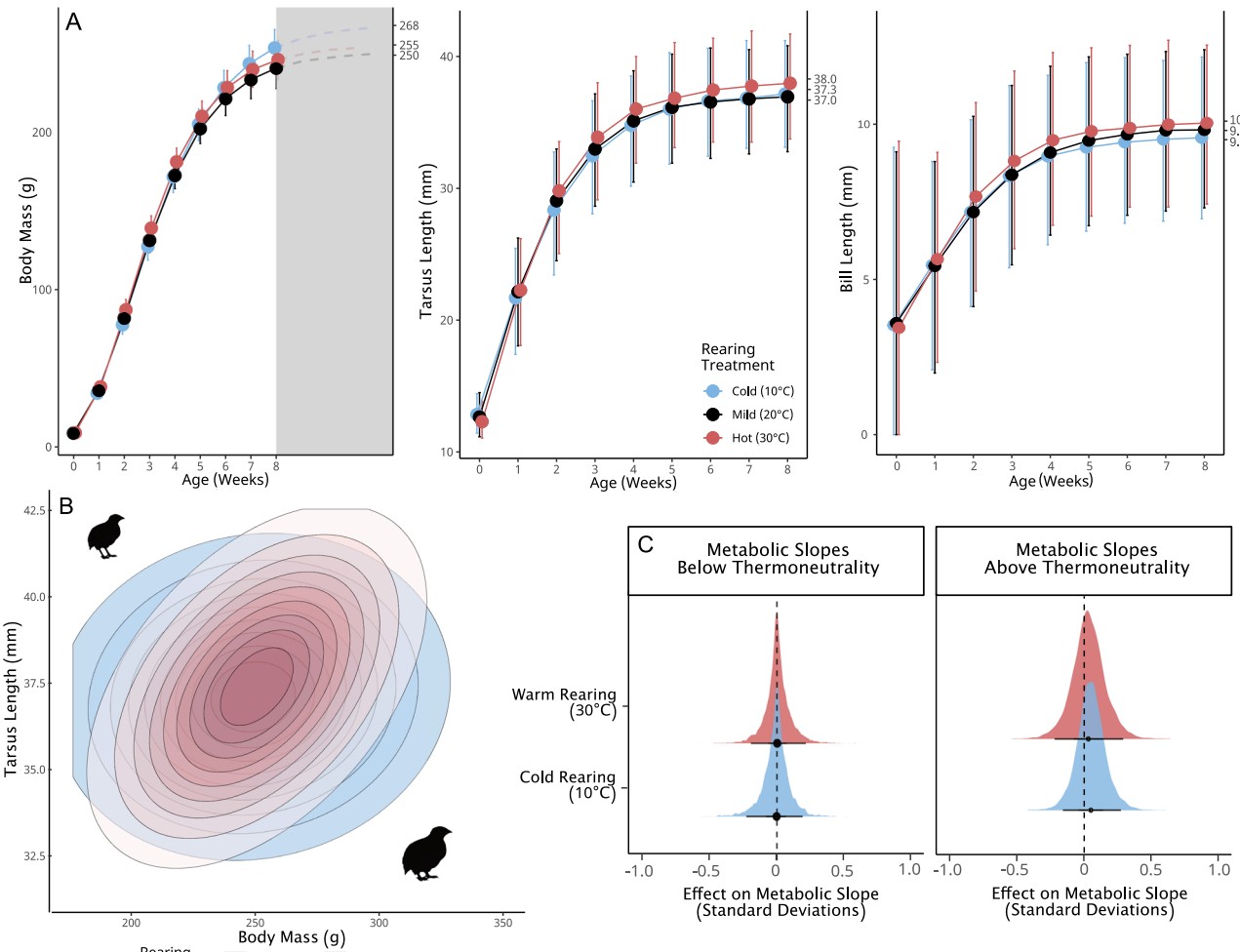

**Fig. 3 | Effects of rearing temperature on growth, tarsus elongation, and thermoregulatory responses to heat and cold at adulthood.** Rearing treatments (cold: n = 43; mild: n = 40; warm: n = 47) were maintained until at least 3 weeks of age. **A** Mass gain and both tarsus and bill elongation across weeks. Large dots display predicted means from a Bayesian Gompertz model, by age, and error bars represent 95% quantile intervals. Asymptotes per treatment are displayed on the right-hand y-axis. In (**A**), the grey rectangle indicates the period of estimated further growth after sexual maturation. **B** Distributions of combined mass and tarsus lengths among adult quail reared in two temperature treatments. Contours show the distribution of phenotypes assuming multivariate normality. **C** The direct effects of plastic differences in mass and tarsus length observed between temperature treatments on subsequent metabolic responses to cold (30–10 °C) and heat (30–40 °C) exposures. Densities represent posterior densities from Bayesian path analyses and the dashed line indicates zero. The quail silhouette was created by J.T.

explain avian shape-shifts[6] is debatable (e.g. see evidence for negative selection on bill length in ref. [46]), but should be investigated further.

Beyond concerns of allometry, evidence from desert birds indicates that size reductions can carry thermoregulatory *costs* in extreme heat waves (~48 °C), by lowering maximal heat tolerance and time to dehydration owing to increased heat flux[21]. Moreover, large-bodied individuals may well have preferential access to ecological resources mediating heat tolerance (e.g. water and forage) compared with smaller-bodied conspecifics. Such factors could help explain continued positive selection on body size in our warming world[23,47]. Evidently, if climate warming is creating new pressures on thermoregulation, whether and how that may change body size and shape in avian populations may be too complex to explain unidirectional shape-shifts reported across species (see ref. [14]).

Finally, our results revealed a tentative, but weak effect of rearing temperature on metabolic responses to heat, at least when rearing was in the warmth. Similar to cold-induced responses, these effects were again more apparent in juveniles than adults and contributed to reductions in metabolic costs among warm-reared quail relative to neutral- and cold-reared quail above thermoneutrality (juveniles; warm rearing: $\beta = -0.017$, BF = 25.059; cold rearing [10 °C]: $\beta = -7.5 \times 10^{-3}$, BF = 4.626; Supplementary Table 69; adults; warm rearing; $\beta = 5.8 \times 10^{-3}$, BF = 2.512; cold rearing [10 °C]:

$\beta = 1.10 \times 10^{-3}$, BF = 1.188; Supplementary Table 79). Such reductions likely signal plastic changes in physiology facilitating evaporative and sensible cooling (known in other bird species; e.g. refs. [39,48,49]). Notably, effects of warm rearing on metabolic responses to heat were broadly comparable to those of relatively large changes in morphology at this stage (Fig. [2]E). As such, any thermoregulatory costs imposed by severely mismatching morphology with expected optima for a given thermal environment (e.g. a large body and short appendages in the heat) could, at least partly, be compensated for by acclimating physiologically to that thermal environment[12].

## Plasticity recapitulates observations of shape-shifting, but with no thermoregulatory benefit

Beyond selection on thermoregulatory response to heat, warming climates may directly alter avian size and shape through neutral, or even non-adaptive phenotypic plasticity[16,50]. To test this, we raised quail in cold (10 °C), mild (20 °C) and warm (30 °C) temperatures, then compared growth rates and both body mass and appendage lengths at maturity.

Supporting a plastic origin of size declines and limb elongations[6,7], quail raised at warm temperatures (n = 47) had smaller asymptotic masses (Fig. [3]A; Δ Gompertz a = −13.08, BF = 113.454; Supplementary Table 13)

but longer asymptotic tarsus and bill lengths (Fig. 3A; tarsus length: Δ Gompertz a = 0.661, BF = 7.954; Supplementary Table 22; bill length: Δ Gompertz a = 0.441, BF = 32.403; Supplementary Table 27) than cold-reared quail (10 °C; n = 46), while growing more quickly (similar to many fish and invertebrates[51]; mass Δ Gompertz c = 0.061, BF > 1000; tarsus length Δ Gompertz c = 0.088, BF = 50.780; bill length Δ Gompertz c = −0.09, BF = 4.036; Supplementary Tables 13, 22 and 27, respectively). These findings largely agree with those of others[39,52], revealing that temperature-dependant developmental plasticity likely does contribute to body size reductions and appendage elongations in the heat relative to the cold (Fig. 3B). Despite this, we found no evidence that emergent morphologies benefited thermoregulation in response to subsequent heat or cold exposures, once effects of physiological acclimation were controlled for. In all cases, plastic changes in morphology led to remarkably weak and uncertain effects on metabolic responses to both heat and cold challenges (Fig. 3C). Thus, temperature-dependent plasticity of size and appendage length does not evidently occur to reduce thermoregulatory costs (i.e. via adaptive plasticity). Instead, heat-induced reductions in mass and increases in appendage length may better reflect the emergent consequences of balancing thermoregulation and growth, and direct effects of temperature on peripheral tissue proliferation. For example, in the warmth, energetic demands of dissipating heat (including that produced from growth) may compete with those of growth, thus shrinking asymptotic masses[50]. Further, stimulating effects of heat on chondrocyte proliferation in developing limbs may also directly increase asymptotic appendage lengths[31,53], but without a subsequent thermoregulatory value.

## Conclusions and future directions

In endotherms, body size and appendage length are widely assumed to influence costs of thermoregulation by affecting surface area to volume ratios, and thus, rates of sensible heat loss. Our results show that this assumption is context-specific and tenuous. In the cold, body size and appendage length had remarkably limited effects on costs of thermoregulation, and only among juveniles (that are poorly insulated). In the heat, both body mass and appendage length independently influenced costs of thermoregulation at all ages. However, phenotypes expected to provide at least moderate energy costs required strong deviations from mass-limb length allometry (i.e. both a large body and short limbs) that were rare (~2.5% of quail). While we cannot rule out the possibility that even small energy costs may be evolutionarily meaningful, these findings appear to suggest that changing thermoregulatory demands from warming climates have limited selective influence on morphology, particularly given the strong and potentially buffering effects of physiological acclimation (via warm- or cold rearing) observed in our birds. If morphological changes observed in contemporary birds are indeed linked to warming climates, neutral temperature-dependent plasticity appears a more likely mechanistic explanation.

Using a controlled experimental approach with liberal, acute temperature challenges, our study provides a comprehensive empirical test of whether and how body size and appendage length may shape thermoregulatory costs in birds. Future studies testing the effect of morphology on thermoregulatory costs in naturalised settings, with longer-lasting heat and cold exposures, will be a critical next step toward determining whether environmental temperature can indeed shape selection on a species' form. Similarly, attempts to determine the precise effects of increased thermoregulatory costs on fitness are equally needed.

## Methods

### Animal husbandry

All animal handling, measurements, and euthanasia for this study were approved by the Malmö/Lund Animal Ethics Committee, acting under authorisation by the Swedish Board of Agriculture (permit no. 9246-19).

Japanese quail eggs ($n_{batch\ 1} = 92$, $n_{batch\ 2} = 60$ and $n_{batch\ 3} = 85$ from 'Jumbo' variety quail) were acquired from a commercial supplier (Sigvard

Månsgård, Åstorp, Sweden), which houses and breeds adults in open-air conditions in temperate, southern Sweden. Exposure of our source population to wide ambient temperature variations (below −5 °C to above 25 °C) is expected to have allowed retention of thermoregulatory physiology similar to that of wild-type quail. Upon acquisition, eggs were held at room temperature and manually or mechanically turned for a maximum of 6 days until incubation in Brinsea OvaEasy 190 incubators (Brinsea, Weston-super-Mare, United Kingdom; calibrated prior to use). Incubation was completed in three asynchronous batches between 2021 and 2022, with temperatures fixed at 37.5 °C and relative humidity (RH) fixed at 50% per batch[25]. In all batches, eggs were shifted to hatching trays within incubators between days 15 and 16 of incubation and monitored twice daily until hatching. Once hatched (n = 47, 45, and 55 for batches one to three; average hatching success including unfertilised eggs ≈63.6%), chicks were collected within 12 h of their earliest possible hatch time, colour banded, measured (see below), and placed into one of three possible animal housing rooms (batches 1 and 2: 12L:12D; batch 3: 14L:10D), each containing identical open pens (310 × 120 × 60 cm) lined with wood shavings. Ground food (turkey starter pellets, [Kalkonfoder Start, Lantmännen, Stockholm, Sweden]), water and crushed seashells were provided *ad libitum* and supplemented with mealworms and mixed shredded vegetables (e.g. lettuce and carrots) haphazardly but equally between pens.

At least 24 h before use, housing rooms were set to one of three ambient temperature treatments (10 °C [cold], 20 °C [mild], or 30 °C [warm]) and monitored daily for temperature deviations. Relative humidity was left at ambient. To aid survivorship before development of endothermy, all pens were equipped with a hanging, infrared heat lamp until chicks reached 2 weeks (experimental batch three) or 3 weeks (refs. 39,54; experimental batches one and two). In batch three, heating from lamps was further restricted by allowing for 6 cooling bouts per day (10 min per 2 h in week 1, and 30 min per 2 h in week 2), with this change accounted for in our statistical analyses (refer to inclusion of batch ID as a group-level effect in our analyses). Food and water were placed away from lamps to require regular departure, thus ensuring consistent exposure to treatment temperatures (see ref. 55). Surface temperatures under lamps averaged ~37.5 °C across pens (determined by infrared thermography or placement of Theromchron iButtons™; OnSolution Pty Ltd., Castle Hill, Australia) and duration of lamp placement did not differ between treatment groups.

At 3 weeks of age, ground feed was altered to lower relative protein density (turkey grower [Kalkonfoder Tillväxt, Lantmännen, Stockholm, Sweden]; 22.5% protein relative to 25.5% in starter feed) and maintained *ad libitum* until study completion. At this time, warm- and cold-reared birds from experimental batches one and two were also shifted to mild conditions (20 °C) until adulthood, as part of another study. All birds reared in warm- or cold-conditions until at least this time are nonetheless considered 'warm-' and 'cold-' reared, respectively, recognising that study outcomes are conservative.

Quail were euthanised upon study completion using inert gas (N₂) followed by destruction of the brain. All experimental groups contained mixed sexes.

### Morphometry measurements

Quail were weighed to the nearest 0.1 g on a digital scale within 12 h of hatching, then weekly until adulthood (8 weeks of age[25]). As indicators of appendage length, we measured the tarsus and bill owing to their long-known importance for avian thermoregulation[56,57]. Bill and tarsus lengths were measured weekly until 3 weeks of age, then again at adulthood (8 weeks). To measure bill length, individuals were flat-lay photographed with a square of 1 × 1 mm grid paper placed over the eye for calibration. Measurement of tarsus length followed a similar method but with the tarsus flat-lay photographed on 1 × 1 mm grid paper, with their left tarsus exposed and digits angled roughly perpendicular to the tarsometatarsus (Supplementary Fig. 5). Digital measurements were used to minimise animal handling time and reduce risk of measurement error reported for analogue measurements[58] (but see ref. 59). Blurred images were removed from

our analyses and means were taken when multiple images were available (41.7% of individuals multiply photographed). From retained images, length measurements were calculated in FIJI[60] as the calibrated straight-line distance between the bill tip and nares (bill length) or between the ankle and the distal end of the tarsometatarsus (tarsus length; matching traditional analogue measurements[58]). Correlations between analogue and digital measurements was confirmed with a subsample (bill length: $\beta = 0.231$ [50% CI: 0.099, 0.360][95% CI: −0.155, 0.638], $R^2 = 0.133$, BF = 7.247, n = 34; tarsus length: $\beta = 0.745$ [50% CI: 0.665, 0.825][95% CI: 0.509, 0.991]; $R^2 = 0.569$; BF > 1000, n = 43; Supplementary Tables 3 and 4). In some cases, calibration paper was laid beside the imaged bill (n = 79; 16.5% of total), above the imaged tarsus (n = 49 images; 10.0% of total) or replaced with a ruler (bill: n = 11; 2.3% of total; tarsus: n = 14 images; 2.9% of total). In these cases, bill and tarsus lengths were adjusted by modelling the effect of calibration type on each length measurements (controlling for categorical weekly age) and subtracting mean effects from estimated bill of tarsus lengths (bill: grid beside: $\beta = 2.099$ [95% CI: 1.905, 2.293]; ruler beside: $\beta = 3.079$ [95% CI: 2.554, 3.606]; tarsus: grid over: $\beta = 3.260$ [95% CI: 2.502, 4.021]; ruler under: $\beta = 2.406$ [95% CI:0.801, 4.004]).

To confirm that body mass predicted structural size in quail, we measured the maximum external body height (the maximal straight-line distance between the base of the keel and spinal dorsum in the transverse plane) and synsacrum width (the maximal distance between the fossa renalii) of a subsample of adults (n = 20), then tested whether body mass predicted these measurements across individuals. Measurements were obtained after euthanasia (~9 weeks of age) and collected using analogue and digital calipers, to the nearest 0.1 mm. To reduce observer biases, all skeletal measurements were collected by two independent researchers (JKRT, EP), and precision of each measurement subsequently confirmed (mean CVs = 1.78% and 2.24% for maximum body depth and synsacrum width, respectively). For both skeletal measurements, Bayesian linear models indicated body mass as a clear predictor (Supplementary Fig. 2.; maximum body height: $\beta = 0.032$ [95% CI: 0.009, 0.054], BF = 194, $R^2 = 0.272$; synsacrum width: $\beta = 0.029$ [95% CI: 0.010, 0.046], BF = 499, $R^2 = 0.341$; priors for main effects normal with mean = 0 and s.d. = 2).

## Respirometry

Temperature-specific, resting metabolic rates were measured once for each quail during development (3–4 weeks; as juveniles) and once during adulthood (8–9 weeks) using flow-through respirometry methods described previously[55]. Here, quail from batches 1–2 were measured at ambient temperatures ranging from 10 °C to 40 °C, while measurements for those in batch three were restricted to 30 °C and 40 °C. Briefly, quail were placed in sealed glass chambers (batches 1 and 2: 3.3 L at 3 weeks and 8.0 L at 8 weeks; batch 3: 13.0 L at all ages), ventilated with dry (drierite, Sigma-Aldrich, Stockholm, Sweden) atmospheric air at least 30 min prior to measurement. To capture faeces and remove its effect on our estimates of evaporative water loss, chambers were supplied with mineral oil reservoirs, over which a metallic mesh grid was placed for quail to stand[55,61]. All chambers were secured within a sealed climate chamber (Weiss Umwelttechnik C180, Reiskirchen, Germany) before bird placement, and the climate chamber was set to 10 °C (batches 1 and 2) or 30 °C (batch 3) for acclimation. Throughout the experiment, air temperature was measured in chambers using thermocouples (36-gauge type T, copper-constantan; thermocouple box: TC-2000, Sable Systems) secured at a position where they were not affected by the birds' heat production. One to four quail were held in our climate chamber at a time for any given set of measurements.

In batches 1 and 2, chamber temperature was sequentially increased by 10 °C increments until 40 °C, with initial baseline (7–15 min), measurement (10 min per bird; totalling 30–40 min), and terminal baseline periods (until stable, and at least 5 min) collected at each temperature (totalling a maximum of 4 measurements per bird and age class). Although the rate of temperature increases are likely to be higher than those experienced by most wild bird species, this method was chosen to align with methods used by others[62,63]. In batch 3, chamber temperatures was increased to 40 °C after a

30–60 min initial measurement period, after which measurements proceeded until gas traces were stable for at least 5 min. Air flow rates across all temperatures averaged 2.2 L min$^{-1}$ ( ± pooled sem = 0.014; 1.9–2.6 L min$^{-1}$; standard temperature and pressure, dry, STDP) for juveniles and 4.2 L min$^{-1}$ STDP (±sem = 0.016; 3.7–4.8 L min$^{-1}$) for adults (measured using a FB8 mass flow metre; Sable Systems, Las Vegas, NV, USA) in batches 1 and 2, from which we subsampled excurrent air at, on average, 351 mL min$^{-1}$ (308–388 mL min$^{-1}$) for analysis. In batch 3, mean flow rates were increased to 10.1 L min$^{-1}$ STDP (±sem = 0.006; 9.5–10.6 L min$^{-1}$) between 30 °C and 40 °C with subsampling averaging 403 mL min$^{-1}$ (390–419 mL min$^{-1}$). In batch 3, flow rate was registered using Alicat 0-20 SLPM flow metres (Alicat Scientific Inc., Tucson, AZ, USA). All subsampling was achieved using a Sable Systems SS-4 subsampler, and 99% equilibration times prior to subsampling ranged from 5.8 to 8.1 min and 7.8 to 10.1 min for juvenile and adult quail, respectively. To measure oxygen and water vapour from subsampled air, we used both an FC-10 oxygen analyser (Sable Systems) and RH-300 water vapour pressure metre (Sable Systems) placed in series, and calibrated as described in ref. 55. Water vapour pressure was measured first, then both water vapour and carbon dioxide were stripped from effluent air (drierite and ascarite II; Acros Organics, Geel, Belgium) and oxygen subsequently measured. All birds showing signs of distress for >5 min (e.g. prolonged gular fluttering or erratic behaviour) were removed from study (n = 5 adults from batch 1 and n = 2 juveniles from batch 2).

To calculate oxygen consumption per bird, we extracted oxygen readings from the most stable 2 min of 10 min recordings (batch 1 and 2) or from the most stable 2 min after gas concentrations had stabilised (batch 3) using the software ExpeData (version 1.9.27; Sable Systems), then converted these to mL O$_2$ min$^{-1}$ following equation 11.1 described in ref. 64. To calculate evaporative cooling efficiency, we first collected water vapour pressure readings from the same 2 min period then converted these to estimates of evaporative heat loss (W) following ref. 64 while assuming 2406J consumed for every 1 mL of water evaporated[65]. Evaporative cooling efficiency then represented the ratio of this evaporative heat loss value to metabolic heat production (estimated from O$_2$ consumption and assuming 20 J = 1 mL O$_2$; ref. 66).

## Data organisation, statistical analyses, and reproducibility

Data organisation and statistical analyses were achieved using R statistical software (version 4.2.3; ref. 67) and the R package brms[68]. Plots were produced using the package ggplot2[69]. Model diagnostics (i.e. Hamiltonian Monte Carlo [HMC] chain diagnostics, prior predictive checks, prior power-scaling sensitivity analyses, posterior predictive checks, residual visualisation, etc.) were achieved visually, guided by ref. 70. For all models, Gelman-Rubin statistics ($\widehat{R}$; ref. 71) exceeded 0.9 and ratios of effective sample sizes to sample sizes ($N_{eff} N^{-1}$) exceeded 0.75, indicating strong chain convergence and little autocorrelation within HMC chains. To minimise bias from skewed posterior distributions, posterior estimates and credible intervals were calculated as medians and quantile intervals respectively, unless otherwise stated. Detailed descriptions of prior derivations and model validations (including R code) are provided in the Supplementary Material, while descriptions of primary analyses are provided in subsections below. All data used in this study are provided in the supplement (Supplementary Dataset 1).

## Calculation of metabolic slopes

Metabolic slopes were calculated as the rate at which an individual increased their resting metabolism from thermoneutrality (30 °C; refs. 55,72) to our lowest temperature exposure (10 °C; ~15 °C below thermoneutrality[72]) or highest temperature exposure (40 °C; ~10 °C above thermoneutrality[72]). Because we were interested in the efficiency with which an individual expended their own energy toward warming, or as consequence of heating, resting metabolism values (mL O$_2$ min$^{-1}$) at all temperatures were first relativised as an individual's fold change from that observed at 30 °C. Doing so allowed us to restrict among-individual variation in metabolism to that

explained by differences in their response to a given temperature (i.e. by fixing resting metabolism at 30 °C [thermoneutrality] for all individuals at 1) rather than also differences in their metabolism at thermoneutrality.

To estimate metabolic slopes in the cold, we used Bayesian linear mixed effects models with relative resting metabolism as our Gaussian-distributed response variable, ambient temperature (°C; 10 °C, 20 °C, and 30 °C, encoded continuously) as the sole population-level predictor, and individual identity as a group-level slope; metabolic slopes per individual then represented their group-level slope of the metabolism by temperature relationship. Separate models were constructed for juveniles and adults. Since, by definition, metabolism values were invariable at 30 °C (equalling 1; see above), we set 30 °C as our x-intercept and fixed our corresponding y-intercepts at 1. Equations for our model were therefore:

$$Relative\ Resting\ Metabolism_{ij} \sim 1 + (\beta_1 + \nu_j) \cdot Ta_{ij} + \varepsilon_{ij}$$

$$Metabolic\ Slope_j \equiv \beta_1 + \nu_j$$

with $\beta_1$ representing the population-level effect of ambient temperature on relative metabolism, $\nu_j$ representing an individuals' deviation from that

conditions were manually calculated from two point measurements, thus precluding estimation of within-individual variation of metabolism across ambient temperature. Importantly, conditional repeatability does not indicate consistency of individual resting metabolism values at any *given* ambient temperature. Rather, conditional repeatability describes the amount of between-individual variation in resting metabolism, relative to within-individual variation, while controlling for ambient temperature. To statistically test our repeatability values against null expectations (which can be greater than 0), we again followed methods described by others[75]. Briefly, models predicting relative metabolism in response to temperature we re-run but while randomly scrambling individual identities among measurements. Repeatabilities calculated from these new models (or 'null models') were then compared against those from our initial models using one-way hypothesis tests (here, using the Savage-Dickey density ratio method[76]). Priors for hypothesis tests are described in the Supplementary Material (page 221).

### Effects of morphology and rearing temperature on metabolic slopes

To partition direct effects of morphology and rearing temperature on metabolic slopes in the cold and heat, we used Bayesian path analyses composed of the following mixed effects models:

$$Body\ mass_j \sim \beta_0 + \beta_1 \cdot Cold\ Rearing_j + \beta_2 \cdot Warm\ Rearing_j + \mu_{0j} + \varepsilon$$
$$Tarsus\ Length_j \sim \beta_0 + \beta_1 \cdot Cold\ Rearing_j + \beta_2 \cdot Warm\ Rearing_j + \beta_3 \cdot Body\ mass_j + \mu_{0j} + \varepsilon$$
$$Bill\ Length_j \sim \beta_0 + \beta_1 \cdot Cold\ Rearing_j + \beta_2 \cdot Warm\ Rearing_j + \beta_3 \cdot Body\ mass_j + \mu_{0j} + \varepsilon$$
$$Metabolic\ Slope_j \sim \beta_0 + \beta_1 \cdot Cold\ Rearing_j + \beta_2 \cdot Warm\ Rearing_j + \beta_3 \cdot Body\ mass_j + \beta_4 \cdot Tarsus\ Length_j + \beta_5 \cdot Bill\ Length_j + \mu_{oj} + \varepsilon$$

population-level effect (here, normally distributed with a mean of 0 and variance $\nu_\nu$), $i$ representing an individual observation, $j$ representing an individual, and $\varepsilon$ representing a normally distributed error term. To capture heterogeneity in relative metabolism observed across ambient temperature (e.g. Supplementary Fig. 72), $\varepsilon$ was allowed to vary between measurement temperatures.

In our adults, thermoneutrality extended below 30 °C, ending at ~24 °C (23.91; BF > 1000; Supplementary Table 113). Thus, to more accurately capture the linear relationship between ambient temperature and relative metabolism at this stage, we adjusted our x-intercept to 24 °C (again, fixing our corresponding y-intercept to 1; linearity confirmed in Supplementary Fig. 83).

Priors for the population-level effect of ambient temperature on relative resting metabolism were informed by refs. 55,73, and set as skew-normal for all ages ($\xi = -0.018$, $\omega = 0.02$, $\alpha = -2.5$; defining the mean near that reported by ref. 73). For the variance around individual slopes ($\nu_\nu$), we used a weak exponential prior ($\lambda = 2.5$), and for that of our error term ($\varepsilon$) at 10 °C (here, natural-log transformed to fix values above 0), we used a weak skew-normal prior ($\xi = -0.2$, $\omega = 1.0$, $\alpha = -5$). Variance is metabolism decreased as temperatures increased. As such, priors for the change in error between 10 °C and 20 °C were normal with a mean above 0 (0.25, SD = 0.25).

In response to heat, metabolic slopes were calculated manually as the change in relative metabolism observed among individual between 40 °C and 30 °C, divided by 10. Manual calculation was done as resting metabolism was only measured at these two temperatures.

### Repeatability of metabolic slopes

To calculate repeatability of metabolic slopes, we followed methods described by others[74]. Here, repeatability values represented conditional repeatabilities (i.e. conditional on, or controlling for, ambient temperature[74]) and were only calculated for responses to cold. In response to heat, calculating repeatability was not possible since metabolic slopes in these

with residual variance between models assumed to be uncorrelated. *Cold Rearing* and *Warm Rearing* were binomial variables encoding whether an individual was reared at 10 °C or 30 °C, respectively (0 equalling 'no' and 1 equalling 'yes'). $\mu_0$ represents a group-level intercept of the egg batch that an individual was derived from. For the majority of models, response variables (and thus $\varepsilon$) were assumed to be normally distributed, and both continuous predictors and response variables were mean-centred to ease interpretation of model intercepts. Among juveniles in the heat, however, variance in metabolic slopes differed widely between egg batches (Supplementary Fig. 119), and was thus corrected by allowing error ($\varepsilon$) to vary by batch. Further, among both juveniles and adults in the heat, some metabolic slopes fell outside of expectations from a normally distributed error but with no evidence of measurement error. Thus, to balance the influence of these individuals, a Student's $t$-distributed error was assumed, centred on zero and with degrees of freedom calculated from our data. In all, four path analyses were constructed: two ultimately predicted metabolic slopes in the cold (one for juveniles and one for adults), and two ultimately predicting metabolic slopes in the heat (again, one for juveniles and one for adults).

For path analyses pertaining to juveniles, priors for the effects of cold and warm rearing on both body mass and appendage length were normal and conservative (body mass: $\bar{x} = 0$, $\sigma = 15$; tarsus length: $\bar{x} = 0$, $\sigma = 2.5$; bill length: $\bar{x} = 0$, $\sigma = 0.5$). Priors for body mass and appendage length intercepts were also normal and conservative (means = 0; SDs = 5, 2.5, and 1 for body mass, tarsus length, and bill length respectively) and those for egg batch effects ($\mu_0$) on, and error ($\varepsilon$) around, body mass and appendage length were exponential and weak (body mass: $\mu_0$: $\lambda = 1.5$, $\varepsilon$: $\lambda = 0.15$; tarsus length: $\mu_0$: $\lambda = 2$, $\varepsilon$: $\lambda = 1$; bill length: $\mu_0$: $\lambda = 5$, $\varepsilon$: $\lambda = 2.5$; derivation described in Supplementary Material, pages 243–274 and 385–390). For path analyses pertaining to adults, priors on predictors of body mass and tarsus length were similar to those for juveniles but broadened to account for increased variance in each variable with age (body mass: intercept = $\mathcal{N}[0, 10]$, cold rearing = $\mathcal{N}[0, 25]$, warm rearing = $\mathcal{N}[0, 25]$, $\mu_0$ = exponential[2.5],

$\varepsilon$ = exponential[0.15]; tarsus length: $\mathcal{N}$ [0, 2.5], cold rearing = $\mathcal{N}$ [0, 2.5], warm rearing = $\mathcal{N}$[0, 2.5], $\mu_0$ = exponential[2], $\varepsilon$ = exponential[1]). Priors on bill length, however, remained the same as those described for juveniles. At all ages, priors for the effects of body mass on tarsus and bill length were skew-normal, following visually-confirmed positive allometry ($\xi = 0$, $\omega = 0.25$, $\alpha = 5$).

For models predicting metabolic slopes in the cold (as part of our path analyses), priors for juveniles and adults were uninformed and as follows: intercepts = $\mathcal{N}$ (0, 0.01), cold rearing = $\mathcal{N}$ (0, 0.025), warm rearing = $\mathcal{N}$ (0, 0.025), mass = $\mathcal{N}$ (0, $1.0 \times 10^{-3}$), tarsus length = $\mathcal{N}$ (0, $2.5 \times 10^{-3}$), bill length = $\mathcal{N}$ (0, $1.0 \times 10^{-3}$), egg batch effects = exponential(50), and error = exponential(10). These priors assumed no previous evidence that each parameter predicted metabolic slopes, and that effects of body mass and appendage lengths greater than the range of metabolic slopes divided by their own ranges were unlikely. For models predicting metabolic slopes in the warmth, uninformed priors were also used and were as follows for juveniles and adults respectively: intercepts = $\mathcal{N}$ (0, 0.125) and $\mathcal{N}$ (0, 0.1), cold rearing = $\mathcal{N}$ = (0, 0.125) and $\mathcal{N}$ (0, 0.1), warm rearing = $\mathcal{N}$ (0, 0.125) and $\mathcal{N}$ (0, 0.1), mass = $\mathcal{N}$ (0, $4.0 \times 10^{-3}$) and $\mathcal{N}$ (0, $2.5 \times 10^{-3}$), tarsus length = $\mathcal{N}$ (0, 0.015) and $\mathcal{N}$ (0, 0.03), bill length = $\mathcal{N}$ (0, 0.06) and $\mathcal{N}$ (0, 0.1), egg batch effects = exponential(50). For juveniles, among which our error term varied by egg batch, error term priors were as follows (again, natural-log transform to fix values above 0): egg batch 1 = $\mathcal{N}$ (-3, 1.5), change from egg batch 1 and egg batch 2 = $\mathcal{N}$ (0, 0.5), and change from egg batch 1 and egg batch 3 = $\mathcal{N}$ (1, 1.5). For adults our prior for the centrality of our error term was exponential (10), and for both ages, that for our degrees of freedom (ν) was conservatively gamma-distributed (juveniles: α = 5, β = 1; adults: α = 10, β = 1).

### Effects of morphology and rearing temperature on evaporative cooling efficiency

Effects of morphology and rearing condition on evaporative cooling efficiency (here, at 40 °C) were again partitioned using Bayesian path analyses. Path analyses followed the same structure as those ultimately predicting metabolic slopes but with the terminal model adjusted as follows:

$$Evaporative\ Cooling\ Efficiency_j \sim \beta_0 + \beta_1 \cdot Cold\ Rearing_j + \beta_2 \cdot Warm\ Rearing_j + \beta_3 \cdot Body\ mass_j + \beta 4 \cdot Tarsus\ Length_j + \beta_5 \cdot Bill\ Length_j + \mu_{0j} + \varepsilon$$

Priors for all predictors of body mass and appendage lengths at each life stage remained identical to those described above. For predictors of evaporative cooling efficiency, priors for juveniles and adults were as follows: intercepts = $\mathcal{N}$ (0.5, 0.2) and $\mathcal{N}$ (0.75, 0.2), cold rearing = $\mathcal{N}$ (0, 0.25), warm rearing = $\mathcal{N}$ (0, 0.25), mass = $\mathcal{N}$ (0, 0.01), tarsus length = $\mathcal{N}$ (0, 0.1), bill length = $\mathcal{N}$ (0, 0.075) and $\mathcal{N}$ (0, 0.15), egg batch effects = exponential(15). Intercept priors were informed by ref. 39 and again, those for body mass, tarsus length, and bill length assumed that effects greater than the range of evaporative cooling efficiency divided by their own ranges were unlikely. Given that error in evaporative cooling efficiency again varied by egg batch among juveniles (Supplementary Fig. 149), priors for our error term at this age class was set as: egg batch 1 = $\mathcal{N}$ (-2, 1), change from egg batch 1 to egg batch 2 = $\mathcal{N}$ (0, 0.5), change from egg batch 1 to egg batch 3 = $\mathcal{N}$ (1, 1). For adults, the prior for our error term was exponential ($\lambda = 5$).

### Effects of rearing temperature on mass gain and appendage elongation

To evaluate whether and how environmental temperature shaped mass gain and tarsus elongation after hatching, we modelled body mass (g), tarsus length (mm), and bill length as Gompertz functions of developmental age, in weeks, from hatching until adulthood (8 weeks). Gompertz parameters ($a$, the asymptote, $b$, the x-axis displacement, and $c$, the growth rate) were then each modelled as functions of cold rearing (binomial; 0 = 'no', 1 = 'yes'), warm rearing (binomial; 0 = 'no', 1 = 'yes'), and egg batch. Here, cold rearing and warm rearing were treated as population-level predictors, and egg batch a group-level intercept. To account for repeated measurements of

individuals across all ages, individual identity was also included as a group-level predictor (here, intercept) of each morphometric measure in addition to Gompertz parameters. Further, to evaluate and adjust for effect of body size scaling on tarsus and bill length measurements, body mass, mean-centred by week of measurement, was also included as a population-level predictor of each appendage length in alongside Gompertz parameters. Our models for body mass, tarsus length, and bill length were therefore as follows:

$$Body\ Mass_{ij} \sim a \cdot e^{-b \cdot e^{-ct}} + \mu_{0j} + \epsilon_{ij}$$

$$Tarsus\ Length_{ij} \sim a \cdot e^{-b \cdot e^{-ct}} + \beta_1 \cdot Mass_{ij} + \mu_{0j} + \epsilon_{ij}$$

$$Bill\ Length_{ij} \sim a \cdot e^{-b \cdot e^{-ct}} + \beta_1 \cdot Mass_{ij} + \mu_{0j} + \epsilon_{ij}$$

where '$t$' represents age in weeks, $\mu_{0j}$ represents a group-level intercept of individual identity, '$Mass$' represents the relative mass of an individual $j$ at a given week ($t$) of measurement relative to the mean at that week, and:

$$a_j, b_j, c_j \sim \beta_0 + \beta_1 \cdot Cold\ Rearing_j + \beta_2 \cdot Warm\ Rearing_j + \mu_{1j}$$

where $\mu_{1j}$ represents the group-level intercepts of egg batch on each Gompertz parameter. Last, because variance in each measure increased natural-logarithmically with age, error terms ($\varepsilon$) were modelled according to the following:

$$\epsilon_{ij} \sim \tau_0 + \tau_1 \cdot ln(t + 1)$$

where $\tau_0$ indicates the error intercept, and $\tau_1$ indicates the rate at which error increased across the natural-log of time + 1.

Priors for our model asymptotes (Gompertz $a$) were informed by data from ref. 39 while those for other Gompertz parameters were informed by refs. 55,77. For our model predicting body mass, priors were as follows: Gompertz $a$ intercept = $\mathcal{N}$ (250, 25), effect of cold rearing on $a$ = $\mathcal{N}$ (7.5, 25), effect of warm rearing on $a$ = $\mathcal{N}$ (−7.5, 25), effect of egg batch on $a$ = exponential(2.5); Gompertz $b$ intercept = $\mathcal{N}$ (3, 1), effects of cold and warm rearing on $b$ = $\mathcal{N}$ (0, 0.5), effect of egg batch on $b$ = exponential(10); Gompertz $c$ intercept = skew-normal (0.5, 0.1, 2.5; assuming no negative growth), effects of cold and warm rearing on $c$ = $\mathcal{N}$ (0, 0.2), effect of egg batch on $c$ = exponential(25); effect of individual identity ($\mu_{0j}$) on mass = exponential(0.5); error term intercept ($\tau_0$; natural-log transformed to fix above 0) = skew-normal(1, 0.5, −10); effect of age on error term $\tau_1$ = skew-normal(1, 0.5, 10). For our models predicting tarsus and bill length, priors remained similar and were, respectively: Gompertz $a$ intercept = $\mathcal{N}$ (37.5, 2.5) and $\mathcal{N}$ (15.7, 2.5), effect of cold rearing on $a$ = $\mathcal{N}$ (−0.35, 2) and $\mathcal{N}$ (−0.22, 1), effect of warm rearing on $a$ = $\mathcal{N}$ (0.35, 2) and $\mathcal{N}$ (0.22, 1), effect of egg batch on $a$ = exponential(1) and exponential(5); Gompertz $b$ intercept = skew-normal(0.6, 0.5, 2.5) and skew-normal(1.7, 0.5, 2.5), effects of cold and warm rearing on $b$ = $\mathcal{N}$ (0, 0.5), effect of egg batch on $b$ = exponential(10); Gompertz $c$ intercept = skew-normal(0.5, 0.1, 2.5; assuming no regression), effects of cold and warm rearing on $c$ = $\mathcal{N}$ (0, 0.25), effect of egg batch on $c$ = exponential(25); effect of scaled body mass = skew-normal(0.5, 0.15, 5); effect of individual identity ($\mu$) = exponential(1.5) and exponential(2.5); error term intercept ($\tau_0$; again, natural-log transformed) = skew-normal(1, 0.5, 10) and skew-normal (1, 0.25, 5); effect of age on error term $\tau_1$ = skew-normal(1, 0.5, 10) and skew-normal(0.2, 0.25, 5).

### Effect size calculation

Where reported, Cohen's D values[78] were calculated as the difference in raw posterior predictions (n = 1000 samples) between comparison states (e.g. mean morphometry and mean morphology + 1 SD), divided by the standard deviation of posterior predictions from our original path analyses.

### Reporting summary

Further information on research design is available in the Nature Portfolio Reporting Summary linked to this article.

## Data availability
All data required to reproduce the results and graphs reported here are provided in the supplement (see Supplementary Dataset 1).

## Code availability
Statistical code used to collate, organise, and analyse our morphometric and physiological data is provided as Supplementary Material.

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

## Acknowledgements

We thank Camilla Björklund for her invaluable assistance with animal care, and Fredrik Andreasson and three anonymous reviewers their helpful comments on an earlier version of this manuscript. Funding for this research was provided by the Wenner-Gren Foundation (UP2021-0038 to JT), the Sven and Lily Lawksi Foundation (20240523 to JT), the Craaford Foundation (20211007 & 20221018 to AN), the Royal Physiographic Society of Lund (2021-41891 to AN, 20221103 to JT), Stiftelsen Lunds Djurskyddsfond, the Lars Hierta Minne (20221121 to JT), and the Swedish Research Council (2020-04686, 2024-05362 to AN). ET was supported by a postdoctoral scholarship from the Carl Trygger Foundation for Scientific Research (CTS21: 1173) and MC was supported by a MSc scholarship from the Gyllenstiernska Krapperup Foundation (KR2022-0046) (both to AN).

## Author contributions

JKRT, AN and EP conceived the study, EP, CÓC, MC, ET and JKRT collected the data, JKRT analysed the data and wrote the manuscript, AN and JKRT conducted primary revisions, and all authors revised the final manuscript.

## Funding

## Competing interests

The authors have no competing interests to report.
