## [Transparent Peer Review file · Communications Biology]

Limited evidence that body size shrinking and shape-shifting alleviate thermoregulatory pressures in a warmer world

Corresponding Author: Dr Joshua Tabh

This manuscript has been previously reviewed at another journal. This document only contains information relating to versions considered at Communications Biology.

Version 0:

Reviewer comments:

Reviewer #1

(Remarks to the Author)

I appreciate the steps the authors have taken to address the concerns I raised in my original review of this paper (Reviewer 1) and the context they provided in their responses. I continue to think that the study is interesting and timely and I think the authors have adequately addressed all of my original concerns, save one.

The lone exception is the decision to not consider bill length (an issue raised by 2 of the 3 original reviewers). It seems like such a clearly relevant thing to consider, that I just don't understand why the authors would not simply run the same analyses with bill length that were run with tarsus?

I appreciate that bills are smaller than tarsi, and they certainly may be less important to thermoregulation in general, but they may also be less allometrically constrained to size and less able to be modulated with respect to heat loss from the appendage through means other than size variation; I think a key take away from McQueen et al. 2023 is that bill length and shape may be more influenced by climate than tarsus as the climate changes. Along those lines, from McQueen's abstract: "Poorer physiological control of heat loss via bird bills likely entails stronger selection for shorter bills in cold climates. This could explain why bird bills show stronger latitudinal size clines than bird legs, with implications for predicting shape-shifting responses to climate change." These things combined suggest to me that bill length is essential to consider.

I don't want to be unreasonable about this - and certainly this research group is at the top of this field and I defer to their expertise and judgement in making the decision to focus on only tarsus length in the paper - and if bill length was not measured or needs to be withheld for some practical reason, I certainly do not think that this needs to keep the manuscript from being published. I also acknowledge that most recent work on size and shape changes have not focused on bill lengths. However, if the data do exist, I would like to at least see those results to assess their impacts on how these results should be interpreted; even if the authors ultimately decide not to include them for some reason in the publication, I think it's important to know if they show dramatically different patterns from the tarsus. I imagine the authors have already checked this and bill and tarsus show similar patterns, but it would be nice to know that if that is the case.

Other than that, I think the authors have addressed my comments effectively and I see no reason this should not be published.

Reviewer #2

(Remarks to the Author)

Thank you for the opportunity to review this manuscript. The manuscript, now entitled "Limited evidence that body size shrinking and shape-shifting alleviate thermoregulatory pressures in a warmer world", provides with an interesting experimental approach in captivity to characterize the thermoregulatory benefits of size reductions and increased limb length in the Japanese Quail (which are likely not providing any significant effect on thermoregulation). This work is quite timely and provides with useful insights into the current topic discussion. I have read a corrected version of the manuscript, with

comments from three different reviewers. I agree with the general opinion that the manuscript is overall well written, with a clear presentation of the results and conclusion, whereas the methodological part may be more difficult to surf. Nonetheless, authors have done a good job editing and preparing this manuscript, responding to R3 concerns. Some personal minor concerns regarding clarity and understandability are provided below.

After reading the corrected manuscript and the reviewers' responses, I have come to some similar concerns as mentioned before, the two main ones being i) the lack of consideration of the bill as a thermoregulatory appendage (as pointed out by R1 and R2), and ii) the length of exposure to the stressful events (R2). Regarding the bill, authors have included a sentence to justify the use of the tarsus length, instead of the bill. This is okay, but I feel is insufficient, as it disregards the possible role of the bill and encourages the use of only one of these appendages in future research. Maybe a more open sentence, and a particular mention of the study species, could prevent disregarding the use of bill length/area in the future (e.g.): "As an indicator of appendage length, we chose to measure tarsi. We did not collect measurements on bill length because tarsi have a relatively larger surface area in the Japanese Quail (3% of total skin surface area in comparison with 0.2% for the bill), and in order to reduce disturbance and human handling of individuals". Regarding the exposure to the heat, I agree with R2 that results may not be comparable to events found in nature (and associated with climate warming), thus suggesting a more cautious approach to the obtained results. This has already been dealt with by the authors. In addition, I provide some minor comments on the corrected manuscript:

Main

L59: You mean thermoregulatory costs? Costs of thermoregulation?

L75: Do you know the thermoneutral zone of the quail? In L478 it is mentioned that some adults are still thermoneutral at 26°C, but what about their maximum end? How far apart do 10 and 40°C fall from the limits of the thermoneutral zone in the quail? Are those comparable?

L82: Authors rear quails at 10 and 30°C. While 10°C is later chosen as the cold spell situation, there is no reference to raising the quails at e.g. 40°C, which would mimic a heat spell. Considering the main text of the manuscript talks about a reduction in body size due to increments in temperature, I find odd not testing the rearing quails in warmer environments.

Results and Discussion

L115: Do you have any of these measures for juvenile individuals that did not reach the adult stage? What was the proportion of these metabolic slopes characteristics among individuals. i.e. what was the prevalence of both "strategies"?

Methods

L324: I am surprised about the low hatching success. Is this normal/comparable to natural populations?

L326: First time I read it, it was not clear to me if each batch was placed in one animal housing room, or if batches were asynchronous (first batch divided in 3 rooms, then second batch divided in 3 rooms after batch 1 experiment ended, etc.). This seems to be clarified in L332, but maybe you could be more specific at the start, maybe a supplementary figure may help here?

L333: In L82 it seems there are only two rearing conditions (10 and 30°C), whereas here you specify three different ones (10, 20 and 30°C). Please rephrase for clarity.

L335: Do you have any information on how this change in lamp provisioning influence survivorship in batch 3, compared to batches 1 and 2? I am a bit concerned about this lamp (but I understand its importance, also from an ethical point of view). Substantially, all nestlings may have been reared at the same temperature (or unknown temperature; if competition leaves certain individuals away from the lamp), during the first two weeks of development. Likely authors do not have access to this data, but for future work it would be interesting to somehow account for behavioural responses (e.g. time spent under the lamp, time spent huddling, etc.) that may have compensated temperature stress, by showing the presence/lack of differences between housing treatments.

L404: It is not clear to me if batches 1 and 2 followed a different protocol than batch 3. I may have not understood properly, but batch number 3 was already at 30°C, so it should have taken less steps to reach 40°C. How did you account for this? Also, in L425 seems to reinforce the fact that you are using two different methods for the different batches. Please rephrase for clarity.

L434: From here you talk about the statistics used in the manuscript. Maybe make the following subdivisions clear to indicate they are under the statistics section.

Version 1:

Reviewer comments:

Reviewer #1

(Remarks to the Author)

The authors have done a great job revising the manuscript, and I think the addition of bill length strengthens the findings. I think the manuscript is acceptable as it is, and I look forward to having this contribution in the literature!

The one semi-substantive observation I had relates to the new bill results (which I think are really interesting. There is a recent paper showing that there is a genetic basis for bill length increases in a species of thrush. It seems like this might fit really nicely with the interpretation of the new bill findings presented here – it seems like maybe this supports the caveat that the authors include that even the small benefit of the bill length increase could result in selection. The paper is Adams et al. 2025. Genetic and morphological shifts in a migratory bird. BMC Biology. Just to be clear, while it seems like this could be interesting context for the bill length result, I don't think it is necessary for the authors to include this (I certainly don't need to

see another revision) but thought it might be relevant, particularly when I was reading the added text at line 246.

Minor editorial catch:

Line 12, "are" seems like it should be "and"

Congratulations again on this excellent work, it was a pleasure to review.

Response to Reviewers

Reviewer #1 (Remarks to the Author):

R1: I appreciate the steps the authors have taken to address the concerns I raised in my original review of this paper (Reviewer 1) and the context they provided in their responses. I continue to think that the study is interesting and timely and I think the authors have adequately addressed all of my original concerns, save one.

Authors: Thank-you for your time and thoughtful comments on our previous manuscript drafts. We are grateful for the clear improvements they have made to this current version.

R1: The lone exception is the decision to not consider bill length (an issue raised by 2 of the 3 original reviewers). It seems like such a clearly relevant thing to consider, that I just don't understand why the authors would not simply run the same analyses with bill length that were run with tarsus?

I appreciate that bills are smaller than tarsi, and they certainly may be less important to thermoregulation in general, but they may also be less allometrically constrained to size and less able to be modulated with respect to heat loss from the appendage through means other than size variation; I think a key take away from McQueen et al. 2023 is that bill length and shape may be more influenced by climate than tarsus as the climate changes. Along those lines, from McQueen's abstract: "Poorer physiological control of heat loss via bird bills likely entails stronger selection for shorter bills in cold climates. This could explain why bird bills show stronger latitudinal size clines than bird legs, with implications for predicting shape-shifting responses to climate change." These things combined suggest to me that bill length is essential to consider.

I don't want to be unreasonable about this - and certainly this research group is at the top of this field and I defer to their expertise and judgement in making the decision to focus on only tarsus length in the paper - and if bill length was not measured or needs to be withheld for some practical reason, I certainly do not think that this needs to keep the manuscript from being published. I also acknowledge that most recent work on size and shape changes have not focused on bill lengths. However, if the data do exist, I would like to at least see those results to assess their impacts on how these results should be interpreted; even if the authors ultimately decide not to include them for some reason in the publication, I think it's important to know if they show dramatically different patterns from the tarsus. I imagine the authors have already checked this and bill and tarsus show similar patterns, but it would be nice to know that if that is the case.

Authors: We appreciate this additional push toward including bill size in our analysis. While formulating our original analyses, inclusion of only one measure of appendage length (in particular, that from the largest bare appendage) seemed both pragmatic and sufficient. However, we understand counter-arguments raised previously and here. Having reconsidered our approach in light of your remarks, we now agree that addition of bill size in our analysis would add to a more comprehensive picture of how appendage length may influence energy consumption in the cold and heat. As such, we have repeated our analyses while including bill length as an additional predictor of: (1) metabolic responses to heat and cold, (2) evaporative responses to heat alone, (3) body temperature responses to heat and cold, and (4) lower critical temperature. Descriptions of bill size measurement and inclusion in our statistical analyses are now provided in our revised methods section (throughout lines 380-401 [measurement and validation], 551-598 [description of metabolic slope models], 608-616 [description of evaporative cooling models] 623-640 [description growth analyses], 665-675 [description of growth analyses]). Outcomes from our updated models have also now been integrated throughout our revised results and discussion (e.g. lines 100-111, 128-140, throughout 201-219, and lines 222-226), depicted

in new figure panels (in particular, Figs. 1C, 1E, 2C, 2E, and 3A), and overviewed in our abstract (lines 10-13).

Inclusion of bill size in our analyses has not changed the general conclusions of our study. To summarise, we now show that, contrasting predictions from biophysical theory (e.g. Allen's Rule), bill size does not influence metabolic responses to cooling (Fig. 1C). Coupled with our findings regarding effects of tarsus length on these metabolic responses (lines 100-106 and 137-140 of the revised manuscript; Fig. 1B of the revised manuscript), this finding further indicates that appendage length may be less important for dictating thermoregulatory efficiency in the cold than expected (particularly among adults). In the heat, bill length, like tarsus length, had a moderate influence on relative metabolic expenditure (lines 201-204 of the revised manuscript), but whole-body phenotypes sufficient to yield even moderate thermoregulatory costs (i.e. Cohen's $D > 0.5$) were still rare (2.5% of our data; refer to lines 209-212 of our revised manuscript). These broad findings are outlined in our revised abstract (refer to lines 10-13 of the revised manuscript). One notable departure from our previous analysis is that bill length was not allometrically constrained in our sample, confirming the conclusions by McQueen et al. 2023 in suggesting that bill length may be more free to respond to selection for thermoregulatory efficiency in the heat than the tarsi. Whether effects of the bill on metabolic responses to the heat are sufficiently large to facilitate such selection is unclear, but we emphasise this possibility in lines 246-252 of the revised manuscript.

R1: Other than that, I think the authors have addressed my comments effectively and I see no reason this should not be published.

Authors: Thank-you again for your comments!

Reviewer #2 (Remarks to the Author):

R2: Thank you for the opportunity to review this manuscript. The manuscript, now entitled “Limited evidence that body size shrinking and shape-shifting alleviate thermoregulatory pressures in a warmer world”, provides with an interesting experimental approach in captivity to characterize the thermoregulatory benefits of size reductions and increased limb length in the Japanese Quail (which are likely not providing any significant effect on thermoregulation). This work is quite timely and provides with useful insights into the current topic discussion. I have read a corrected version of the manuscript, with comments from three different reviewers. I agree with the general opinion that the manuscript is overall well written, with a clear presentation of the results and conclusion, whereas the methodological part may be more difficult to surf. Nonetheless, authors have done a good job editing and preparing this manuscript, responding to R3 concerns. Some personal minor concerns regarding clarity and understandability are provided below.

Authors: Thank-you for this feedback. In responding to your comments below, we hope that the methods section is now more readable and clear.

R2: After reading the corrected manuscript and the reviewers’ responses, I have come to some similar concerns as mentioned before, the two main ones being i) the lack of consideration of the bill as a thermoregulatory appendage (as pointed out by R1 and R2), and ii) the length of exposure to the stressful events (R2). Regarding the bill, authors have included a sentence to justify the use of the tarsus length, instead of the bill. This is okay, but I feel is insufficient, as it disregards the possible role of the bill and encourages the use of only one of these appendages in future research. Maybe a more open sentence, and a particular mention of the study species, could prevent disregarding the use of bill length/area in the future (e.g.): “As an indicator of appendage length, we chose to measure tarsi. We did not collect measurements on bill length because tarsi have a relatively larger surface area in the Japanese Quail (3% of total skin surface area in comparison with 0.2% for the bill), and in order to reduce disturbance and human handling of individuals”. Regarding the exposure to the heat, I agree with R2 that results may not be comparable to events found in nature (and associated with climate warming), thus suggesting a more cautious approach to the obtained results. This has already been dealt with by the authors.

Authors: Thank-you for raising the possibility of including bill size in our analysis again. We appreciate the final push and have now chosen to do as requested. For a comprehensive explanation of corresponding changes to the manuscript, please refer to our response to reviewer one above. Briefly, descriptions of bill size measurement and inclusion in our statistical analyses are now provided in our revised methods section (lines 380-401 [measurement and validation], throughout 551-598 [description of metabolic slope models], throughout 608-616 [description of evaporative cooling models] throughout 623-640 [description growth analyses], 665-675 [description of growth analyses]). Outcomes from our updated models have also now been integrated throughout our revised results and discussion (e.g. lines 100-111, 128-140, throughout 201-219, and lines 222-226), depicted in new figure panels (in particular, Figs. 1C, 1E, 2C, 2E, and 3A), and overviewed in our abstract (lines 10-13).

As stated to Reviewer 1, inclusion of bill size in our analyses has not substantively changed the conclusions of our study. Nevertheless, we think that this addition has both strengthened and broadened our study’s findings. To summarise shortly, bill size in our quail did not influence metabolic responses to cooling (Fig. 1C), contrasting predictions from theory (i.e. Allen’s Rule). This finding, coupled with no- or age-specific effects of tarsus length on these metabolic responses (lines 100-106 and 137-140 of

the revised manuscript; Fig. 1B) of the revised manuscript), indicates that appendage length may well be less important for setting thermoregulatory efficiency in the cold than expected (particularly among adults). In the heat, bill length had a moderate influence on relative metabolic expenditure (lines 201-204 of the revised manuscript; Fig. 2C), similar to tarsus length, but whole-body phenotypes sufficient to impose at least moderate thermoregulatory costs (i.e. Cohen's $D > 0.5$) were exceedingly rare (2.5% of our data; refer to lines 209-212 of our revised manuscript). One notable addition to our study is that bill length was not allometrically constrained in our quail, indicating that bill length may be more free to respond to selection for thermoregulatory efficiency in the heat than the tarsi (depending on real-world benefits of bill length on metabolic responses to heat, which appear small here). This finding is described in lines 246-252 of the revised manuscript.

R2: In addition, I provide some minor comments on the corrected manuscript:

R2 - L59: You mean thermoregulatory costs? Costs of thermoregulation?

Authors: Thank-you for catching this error. “[T]hermoregulatory” has now been changed to “thermoregulation” as suggested (refer to line 59 of the revised manuscript).

R2 - L75: Do you know the thermoneutral zone of the quail? In L478 it is mentioned that some adults are still thermoneutral at 26°C, but what about their maximum end? How far apart do 10 and 40°C fall from the limits of the thermoneutral zone in the quail? Are those comparable?

Authors: This is an insightful question. Previous studies have placed the lower critical temperature of Japanese quail around 25°C (Ben-Hamo et al, 2010), which is similar to that in our population (see line 509 of the revised manuscript). For the upper critical temperature, estimates suggest that it likely lays between 30°C (Ben-Hamo et al, 2010) and 38°C (Weathers, 1981). Given the rise in metabolism observed between 30°C and 40°C among our adults, we expect our upper critical temperature to also fall within that range.

In our revised manuscript, we have added tentative distances from thermoneutrality for our 10°C and 40°C exposures, assuming lower and upper critical temperatures reported previously by others (Ben-Hamo et al, 2010; refer to lines 481-484). Although these distances are not equal, we do not expect that this inequality should bias outcomes of our results given metabolic responses below and above these critical values should be linear. Moreover, since the physiological systems engaged to cope with temperatures below and above thermoneutrality are distinct, we question whether drawing comparisons at equivalent distances from critical temperatures is meaningful.

R2 - L82: Authors rear quails at 10 and 30°C. While 10°C is later chosen as the cold spell situation, there is no reference to raising the quails at e.g. 40°C, which would mimic a heat spell. Considering the main text of the manuscript talks about a reduction in body size due to increments in temperature, I find odd not testing the rearing quails in warmer environments.

Authors: We agree that raising quail at 40°C would lend to a nice experimental design, but since Japanese quail show a marked increase in respiratory evaporative water loss already at 35°C, and have a maximal upper temperature tolerance around 45°C (Persson, E., Correia, M., Nord, A., unpublished data), we have reason to suspect that rearing of quail at a constant 40°C is ethically dubious and may reduce survivorship. Even while housing at 30°C, our birds regularly displayed gular fluttering, even before maturation. At 40°C, evaporative water loss increased by approximately 130% among juveniles and 120% among adults relative to themselves at 30°C, indicating a clear exacerbation of this fluttering and, likely, the addition of panting.

R2 - L115: Do you have any of these measures for juvenile individuals that did not reach the adult stage? What was the proportion of these metabolic slopes characteristics among individuals. i.e. what was the prevalence of both “strategies”?

Authors: We have no reason to believe that the individuals failing to reach adulthood did so because of heat- or cold stress. It is simply an unfortunate fact that, even in a controlled, captive setting, some individuals are lost due to random mortality events. That said, we agree that the question raised is conceptually intriguing. Unfortunately, we do not have these types of measures, although we agree that they would be interesting to know. With respect to strategies, we presume this refers to whether the slopes are consistent or inconsistent among individuals. Given that repeatability is an inclusive metric for the entire population, we are unable to divide individuals into distinct “consistency” categories without further analysis. However, we fully agree that this avenue should be explored in future studies.

R2 - L324: I am surprised about the low hatching success. Is this normal/comparable to natural populations?

Authors: In wild Japanese quail, hatching success had been reported at approximately 77% (Chang et al, 2009). This value, however, appears to be as low as 22% in domesticated strains (Chang et al, 2009). From our experience hatching quail obtained from several source populations, these particular hatching success values were not unusual. Importantly, given outcomes of candling, low hatching was probably driven by low fertility rather than poor hatchability. In line 347 of the revised manuscript, we now explicitly state that hatching success values include unfertilised eggs in estimation.

R2 - L326: First time I read it, it was not clear to me if each batch was placed in one animal housing room, or if batches were asynchronous (first batch divided in 3 rooms, then second batch divided in 3 rooms after batch 1 experiment ended, etc.). This seems to be clarified in L332, but maybe you could be more specific at the start, maybe a supplementary figure may help here?

Authors: We appreciate raising the term “asynchronous” here, and have now inserted it into line 344 of the revised manuscript to clarify our design.

R2 - L333: In L82 it seems there are only two rearing conditions (10 and 30°C), whereas here you specify three different ones (10, 20 and 30°C). Please rephrase for clarity.

Authors: Thank-you for this. We have now added mention of our 20°C treatment in line 82 of the revised manuscript as requested.

R2 - L335: Do you have any information on how this change in lamp provisioning influence survivorship in batch 3, compared to batches 1 and 2? I am a bit concerned about this lamp (but I understand its importance, also from an ethical point of view). Substantially, all nestlings may have been reared at the same temperature (or unknown temperature; if competition leaves certain individuals away from the lamp), during the first two weeks of development. Likely authors do not have access to this data, but for future work it would be interesting to somehow account for behavioural responses (e.g. time spent under the lamp, time spent huddling, etc.) that may have compensated temperature stress, by showing the presence/lack of differences between housing treatments.

Authors: We appreciate the concern over these differences in batch treatment. While further restriction of heat lamp access in batch three may well have contributed to slightly different degrees of

temperature-dependent phenotypic plasticity relative to batches 1 and 2, we have sought to correct for these differences by including batch ID as a group-level intercept in our statistical analyses. This correction is now explicitly stated in a parenthetical statement spanning lines 361-362 of the revised manuscript. Even if chicks in the different batches made differential use of the heat lamp before competent endothermy was developed, our previous work shows that any developmental priming of heat tolerance is plastic in entity (Persson et al. J. Exp. Biol. 2024). Thus, the developmental environment per se is unlikely to have influenced the outcomes of measurements in adults.

R2 - L404: It is not clear to me if batches 1 and 2 followed a different protocol than batch 3. I may have not understood properly, but batch number 3 was already at 30°C, so it should have taken less steps to reach 40°C. How did you account for this? Also, in L425 seems to reinforce the fact that you are using two different methods for the different batches. Please rephrase for clarity.

Authors: Thank-you for the suggestion. We have now added a statement to lines 420-422 in the revised manuscript that explicitly states the different temperature ranges of measurement used for batches 1-2 and batch 3. Further, protocols for temperature ramping and respirometric measurement are now outlined per batch (lines 434-440, with lines 439-440 being new).

R2 - L434: From here you talk about the statistics used in the manuscript. Maybe make the following subdivisions clear to indicate they are under the statistics section.

Authors: Thank-you for raising this ambiguity. We have now added section and subsection numbers throughout our manuscript to clarify which paragraphs fall under which category.

Response to Reviewers

Reviewer #1 (Remarks to the Author):

Reviewer 1: The authors have done a great job revising the manuscript, and I think the addition of bill length strengthens the findings. I think the manuscript is acceptable as it is, and I look forward to having this contribution in the literature!

The one semi-substantive observation I had relates to the new bill results (which I think are really interesting). There is a recent paper showing that there is a genetic basis for bill length increases in a species of thrush. It seems like this might fit really nicely with the interpretation of the new bill findings presented here – it seems like maybe this supports the caveat that the authors include that even the small benefit of the bill length increase could result in selection. The paper is Adams et al. 2025. Genetic and morphological shifts in a migratory bird. BMC Biology. Just to be clear, while it seems like this could be interesting context for the bill length result, I don't think it is necessary for the authors to include this (I certainly don't need to see another revision) but thought it might be relevant, particularly when I was reading the added text at line 246.

Authors: Thank-you again for the time reviewing our manuscript and for thoughtful comments. We are grateful for the improvements they have brought to our study.

Adams et al (2025) is a highly relevant article with respect to ours and we're glad to have it raised here. We have now included direct mention of it in our revised manuscript (line 241).

Reviewer 1: Minor editorial catch: Line 12, “are” seems like it should be “and”

Authors: Thank-you for catching this. “[A]re” has now been replaced with “and” in line 10 of our revised manuscript.